# Tuning Electro-Magnetic Interference Shielding Efficiency of Customized Polyurethane Composite Foams Taking Advantage of rGO/Fe_3_O_4_ Hybrid Nanocomposites

**DOI:** 10.3390/nano12162805

**Published:** 2022-08-16

**Authors:** Hussein Oraby, Hesham Ramzy Tantawy, Miguel A. Correa-Duarte, Mohammad Darwish, Amir Elsaidy, Ibrahim Naeem, Magdy H. Senna

**Affiliations:** 1Department of Chemical Engineering, Military Technical College, Cairo 1111, Egypt; 2Centro de Investigacions Biomedicas (CINBIO), Universidade de Vigo, 36310 Vigo, Spain; 3Department of Radar, Military Technical College, Cairo 4393010, Egypt; 4Radiation Chemistry Department, National Center for Radiation Research and Technology, Atomic Energy Authority, Cairo 11762, Egypt

**Keywords:** radiation interference, shielding, polyurethane foam, magnetite decorated rGO, mechanical optimization

## Abstract

Electromagnetic interference (EMI) has been recognized as a new sort of pollution and can be considered as the direct interference of electromagnetic waves among electronic equipment that frequently affects their typical efficiency. As a result, shielding the electronics from this interfering radiation has been addressed as critical issue of great interest. In this study, different hybrid nanocomposites consisting of magnetite nanoparticles (Fe_3_O_4_) and reduced graphene oxide (rGO) as (conductive/magnetic) fillers, taking into account different rGO mass ratios, were synthesized and characterized by XRD, Raman spectroscopy, TEM and their magnetic properties were assessed via VSM. The acquired fillers were encapsulated in the polyurethane foam matrix with different loading percentages (wt%) to evaluate their role in EMI shielding. Moreover, their structure, morphology, and thermal stability were investigated by SEM, FTIR, and TGA, respectively. In addition, the impact of filler loading on their final mechanical properties was determined. The obtained results revealed that the Fe_3_O_4_@rGO composites displayed superparamagnetic behavior and acceptable electrical conductivity value. The performance assessment of the conducting Fe_3_O_4_@rGO/PU composite foams in EMI shielding efficiency (SE) was investigated at the X-band (8–12) GHz, and interestingly, an optimized value of SE −33 dBw was achieved with Fe_3_O_4_@rGO at a 80:20 wt% ratio and 35 wt% filler loading in the final effective PU matrix. Thus, this study sheds light on a novel optimization strategy for electromagnetic shielding, taking into account conducting new materials with variable filler loading, composition ratio, and mechanical properties in such a way as to open the door for achieving a remarkable SE.

## 1. Introduction

In recent years, electronics and technology have grown at a breakneck pace. Furthermore, the smart village models are fast gaining traction, leading to a rise in sensor coverage and wireless communications [1]. This fast expansion of electronics has driven a new sort of pollution known as electromagnetic interference (EMI), which seems to be the direct interference of electromagnetic waves among electronic equipment that frequently affects their normal efficiency. Many concerns have been raised to overcome this issue and protect equipment against these types of interferences. In this context, protecting or shielding the electronics from interfering radiation, which is commonly accomplished by using a shield substance, is one technique to alleviate the problem of EMI. The three basic processes by which a shield substance can attenuate interfering electromagnetic radiation are reflection (R), absorption (A), and multiple reflections (MR) [2]. The performance of the shield material is evaluated in terms of its total shielding effectiveness (SE_T_), which is the totality of losses caused by all three approaches, as shown in Equation (1):SE_T_ = R + A + MR(1)

Traditionally, metals have been used for EMI shielding. However, they display several drawbacks/or suffer from some limitations such as proclivity for corrosion, a difficult manufacturing approach, a large density, and even a shielding approach that is dominated by reflection. In this regard, electrical conductivity (σ), magnetic permeability (μ), dielectric permittivity (ε), and shield thickness (t) are all specific parameters that influence the effectiveness of EMI shielding [3,4]. The link between the material characteristics of the three principal shielding processes is shown in Equations (2)–(4). Depending on these equations, it can be inferred that no individual component will possess all of the required characteristics for acting as an effective EMI shield that absorbs EM radiation [1].
R = 20 log_10_ [0.0997 (σ/f μ_r_ ε_o_) ^0.5^(2)
A = 0.017258t (f μ_r_ σ) ^0.5^(3)
MR = 20log [1 − 10^−(A/10)^](4)

It has been noted that to achieve optimal absorptive losses, an effective EMI shield should maintain high magnetic permeability and electrical conductivity. Therefore, accordingly, the current strategy in shielding enhancement is to combine specific materials with these critical and effective properties [5,6,7], for instance, polymer foam-based composites. These composites are widely used in these applications due to their lower density, corrosion and wear resistance, competitive prices, and high flexibility. On the other hand, the majority of polymer foams are considered as electrical insulators [7,8]. To overcome these limitations and enhance the electrical conductivity of the polymer matrix, a conductive filler can be an effective candidate in such a way as to improve the global conductivity of these compositions. In this sense, a wide range of metallic fillers (e.g., Ag, Au, Cu nanoparticles) [9,10] and carbon-based fillers (e.g., carbon black, carbon nanofibers, carbon nanotubes, and graphene) have been extensively utilized [11,12]. It has been reported that using carbon nanofillers to improve the polymer foam’s conductivity is highly efficient compared to using metallic fillers. This can be attributed to their outstanding characteristics such as higher conductivity, great tensile strength, Young’s modulus, fewer corrosion problems, and lower density [13]. In addition, graphene is widely employed as a filler in different matrices for EMI SE applications. Graphene is a two-dimensional nanomaterial with remarkable electrical characteristics, making it a promising candidate for EMI shielding [14]. Moreover, chemically reduced graphene oxide (rGO) and thermally exfoliated graphene (TEG) have been investigated for EMI shielding, individually or in conjunction with other conducting or magnetic substances [15,16].

Particularly, it has been noted that the developed polymer composites should present not only moderate conductivity and high permeability, but also great mechanical behavior for effective EM wave absorption, and this can be accomplished by mixing conductive and magnetic fillers into the final polymer matrix [17,18]. It has been stated that increasing the filler (conductive/magnetic) loading in the matrix can enhance the shielding efficiency as well as improve the mechanical properties of the shielding material to a certain extent, and a further increase in the loading value can display a negative impact on its mechanical characteristics [19,20]. Therefore, an optimization study among the filler loading, SE, and mechanical properties is essential to reflect a real picture of the optimum filler loading. This optimization study highlights the total profile picture of the prepared shielding substance to gain an acceptable SE and satisfactory mechanical properties to be efficiently applied in the market. In this context, many researchers have addressed and discussed the impact of (Fe_3_O_4_/graphene) as a hybrid nanofillers on EMI SE applications. For instance, Guolong et al. reported a comparative study between graphene nanoplates (GN_S_)/Fe_3_O_4_ and multi-wall carbon nanotubes (MWCNTs)/Fe_3_O_4_ as hybrid fillers in the PU matrix for EMI SE applications. The study revealed that the impact of the MWCNTs on SE was better than GN_S_ due to the one-dimensional geometrical shape of the MWCNTs. Moreover, the addition of Fe_3_O_4_ can effectively enhance the SE in GN_S_ other than in the MWCNTs and this can be attributed to the two-dimensional geometrical shape of GN_S_ enhancing the attraction between Fe_3_O_4_ and decreasing the formation of GN_S_ aggregates. Although the study discussed the impact of these different nanofillers on the EMI SE performance, there is a lack of/scarce information on the effect of varying Fe_3_O_4_ percentage in both composites and its impact on EMI SE [14]. In a different study, Sung et al. developed rGO decorated with Fe_3_O_4_ as a filler in the PU matrix for the EMI SE applications. The study discussed the orientation of the Fe_3_O_4_@rGO composite in the in-plane and out-of-plane by applying an external magnetic field. They concluded that the in-plane orientation enhanced SE by 250% from its initial value. Furthermore, this study did not offer any detailed information about the impact of changing the concentration of Fe_3_O_4_ in the prepared composite on EMI SE [20]. Varsha et al. fabricated a low-density magnetic polymer foam containing a variable loading mass of Fe_3_O_4_ for EMI SE applications. They reported that the addition of Fe_3_O_4_ enhanced the microwave absorption by 82%. However, they only studied the reflection loss and there was no clear information on the transmission loss to show the right picture about the profile/attitude of the developed shielding material [21]. Qing et al. prepared a composite based on epoxy silicone resin coated with carbonyl iron flakes and applied it in EMI SE applications. The results indicated that they gained a minimum reflection loss peak of 42.5 dB at 10.6 GHz for the sample with 55 wt% of iron carbonyl. However, there have been no reports on the transmission loss values/results and no clear performance assessment of the shielding material (reflection loss) over the total band range (2–18 GHz) [22]. Based on the previous discussions, it is clear that there is only scarce information about the optimization study between the filler concentration, SE, and mechanical characteristics of the shielding material. Therefore, in this work, various hybrid nanocomposites based on rGO decorated with different concentrations of Fe_3_O_4_ nanoparticles were synthesized by the co-precipitation method and successfully characterized by transmission electron spectroscopy, X-ray diffraction, Raman spectroscopy, and scanning electron microscopy. The developed Fe_3_O_4_@rGO composites were used as (conductive/magnetic) fillers with different loading (wt%) in the PU matrix to fabricate a (Fe_3_O_4_@rGO/PU) system and further evaluate their performance in EMI SE applications. Finally, this study introduces a novel optimization study among filler loading wt% and the mechanical properties of the developed foam composites for effective electromagnetic shielding applications.

## 2. Experimental Work

### 2.1. Chemicals

All of the reagents used in the preparation of the rGO, Fe_3_O_4_@rGO, and Fe_3_O_4_@rGO/PU composite foams are listed in Table 1. All chemicals were used as obtained without additional purification.

### 2.2. Synthesis of rGO and Fe_3_O_4_@rGO Hybrid Nanocomposites

RGO was synthesized via the modified Hummers’ method (R) as discussed elsewhere [23,24]. The Fe_3_O_4_@rGO composites were synthesized by the co-precipitation method.

Briefly, a solution of (NH_4_)_2_ Fe (SO_4_)_2_ 6H_2_O and FeCl_3_ (with a 1:2 molar ratio) were dissolved in 25 mL of deionized (DI) water. The solution was then dropped into 1000 mL of 1.5 mol/L NaOH/rGO with different rGO wt% (5, 10, 20, and 30)% solutions while being vigorously magnetically stirred at 120 °C. Magnetic separation was used to collect the black precipitates, which were then washed three times with DI water and ethanol. The resulting composites were then dried in a vacuum oven overnight at 60 °C [25]. The acquired rGO coated with Fe_3_O_4_ were labeled according to wt% of rGO as follows Fe_3_O_4_, MR5, MR10, MR20, and MR30.

### 2.3. Synthesis of Fe_3_O_4_@rGO/PU Composite Foams

Four distinct nanocomposite foam samples were developed by dispersing 35 wt% (6.4 g) from each of the prepared MR5, MR10, MR20, and MR30 in the polyol mixture for 10 min at 50% amplitude using a 576-watt probe sonicator (model PRO-250, Hoverlabs, New York, NY, USA). After MDI was added to the polyol mixture at a weight ratio of 1.2:1.0 (6.7 g:5.3 g), the components were vigorously mixed in a 300 mL cup using a wood stick. The mixture was allowed to cure for 24 h at 25 °C [19]. The synthesized samples were labeled as follows: PU (neat), PMR5, PMR10, PMR20, and PMR30, according to the filler composition. The previous method was repeated, but this time by changing the concentration (wt%) of MR20 as a filler in the PU matrix. Three distinct nanocomposite foam samples were synthesized by dispersing different concentrations of MR20 (15, 25, and 35%) wt% (2.2, 4.2 and 6.4) g in the polyol mixture. The fabricated composite foams were labeled as follows: P_1_MR20, P_2_MR20, and P_3_MR20, respectively, according to the filler concentration.

### 2.4. Characterization

Transmission electron microscopy (TEM) images were captured using a JEOL JEM 1010 transmission electron microscope operating at an acceleration voltage of 100 kV equipped with a CCD camera. For the TEM analysis, the samples were acquired by dropping a diluted suspension of the samples onto an ultrathin carbon-coated copper grid.

The morphologies of the obtained samples were studied using scanning electron microscopy (SEM) (type Zeiss EVO-10, Berlin, Germany), equipped with an EDX model (Bruker Nano GmbH 410-M, Berlin, Germany). Polymeric foam pieces were cryo-fractured using liquid nitrogen to obtain the pure undeformed samples for the measurements. X-ray diffraction (XRD) spectra were collected using an X-ray diffraction (XRD) model Siemens D5000 powder X-ray diffractometer (Siemens, Houston, TX, USA) (Cu Ka radiation (λ = 1.54056 Å). Data were recorded in the range (2θ = 5–100°) with an angular step size of 0.026° and a counting time of 0.4 s per step throughout a temperature range of 4–80 °C. Raman spectra were collected at a resolution of 4 cm^−1^ with dispersive Raman spectroscopy (model Renishaw, Gloucestershire, UK). A Nikon 20× objective lens and a neodymium-doped yttrium aluminum garnet (Nd: YAG) excitation source with a wavelength of 532 nm and a power of 10 mW was employed to focus the laser beam.

Fourier transform infrared analysis (FTIR) was used to study the nature of the interactions (physical/chemical) between the Fe_3_O_4_@rGO and the PU matrix. The JASKO 4100 spectrometer (Japan) (Jasko, Tokyo, Japan) was utilized to perform FTIR analysis with a resolution of 4 cm^−1^ in the 400–4000 cm^−1^ range. With each sample, combining 100 scans gave the exact spectra (with a high signal-to-noise ratio).

The magnetic properties (saturation magnetization M_s_, magnetic remanence M_r_, and coercivity H_c_) of the magnetite nanoparticles, MR5, MR10, MR20, and MR30 were evaluated at room temperature using the vibrating sample magnetometer (VSM) model Hielscher, UP200S, (Hielscher, Hamburg, Germany).

The thermal stability of the composite foam materials was investigated using the thermogravimetric method (TGA) (model TGA-60, Shimadzu, Kyoto, Japan) supported by an inert environment and exceeded a temperature rate of 10 °C/min from ambient to 600 °C.

The compression assessments were conducted according to ASTM D 1621-94 via universal testing equipment (model UE2201, Laryee Technology, Beijing, China) to physically test both the pristine and loaded PU foam specimens in compression mode. Samples were prepared for the test by cutting out a cross-sectional area of 25.8 cm^2^ and a height of 2.54 cm. When stress is applied to the specimen at a strain rate of 2.5 mm/min, the stress–strain curve is instantly recorded. The compressive modulus was assigned to the slope of the initial linear part of the curve, whereas the compressive strength was applied to the value immediately before the start of the plateau region. Three samples were analyzed for each sample, and the average value was calculated.

The electrical conductivities of the neat PU and Fe_3_O_4_@rGO/PU composite foam samples were measured according to ASTM D257. A voltage in the range of 0.1 to 1 V was provided to the four-probe electrode model Keithley 220 using a power source (Zahner mess Technic, Model IM6ex potentiostat, Gundelsdorf, Germany) and the specimen under testing (dimensions: 5 × 15 × 15 mm^3^). For every specimen, three measurements were recorded, and the average conductivity value was calculated.

The shielding effectiveness of the manufactured composite foam specimens was tested over the frequency band of 8–12 GHz using a vector network analyzer (VNA model R & S^®^ ZVA-24, Rhode and Shwarz, Columbia, MD, USA). For instrumentation purposes, a rectangular specimen with dimensions of 22.86 × 10.16 mm was made to be fitted in an aluminum alloy waveguide (WR-90) to be measured. The network analyzer measures the S-parameters of the specimen under test (S_11_, S_12_, S_22_, and S_21_). We can compute the total shielding (SE_Total_), reflection loss (RL), and transmission loss (TL) using the measured S-parameters. Our earlier study covered all of the comprehensive theoretical calculations and theory of the EMI SE [19]. Every sample was tested three times to produce an average value of SE, and the average value was recorded. A full two-port calibration was executed before every test.

## 3. Results and Discussion

Clusters of iron oxide (Fe_3_O_4_/γ-Fe_2_O_3_) nanoparticles were acquired via a co-precipitation method, taking advantage of a mechanism in one step as follows:(5)$${\mathrm{Fe}}^{2+}9\left(\mathrm{aq}\right)+2{\mathrm{Fe}}^{3+}\left(\mathrm{aq}\right)+8{\left(\mathrm{OH}\right)}^{-}\overset{}{\to }{\mathrm{Fe}}_{3}O_{4}+4H_{2}O$$

Hybrid nanocomposites were synthesized through the co-precipitation method by mixing the Fe^+2^/Fe^+3^solution in a basic medium at 120 °C. Figure 1 includes the representative TEM images of these clusters of iron oxide nanoparticles, rGO, and of the iron oxide/reduced graphene oxide hybrid nanocomposites with different rGO wt%, coded as MR5, MR10, MR20, and MR30.

Aside from showing the laminar nature of the reduced graphene oxide sheets (Figure 1b), the TEM micrograph included in Figure 1a emphasizes the formation of iron oxide nanoparticles grouped in clusters of controlled size, as confirmed by the size distribution analysis (inset), showing an average diameter of 74 ± 20 nm (Gaussian fitting).

In addition, the TEM images in Figure 1c–f emphasize the successful formation of different hybrid nanocomposites with a uniform distribution of magnetite nanoparticles over rGO sheets and consequently, suppress their unwanted agglomerations [26].

Figure 2 demonstrates the structural characterization of the synthesized samples of the Fe_3_O_4_ cluster nanoparticles, rGO, and MR5, MR10, MR20, and MR30, taking into consideration the X-ray diffraction (XRD). The diffractogram of rGO displayed a broad peak at 2θ = 25° which corresponded to the (002) plane and at the same time, its intensity showed an upward trend, achieving its maximum for the highest rGO loading wt% (MR30), while on the other hand, it had no sense for pure Fe_3_O_4_ [27,28]. Additionally, the XRD analysis of these samples displayed well-defined diffracted peaks that corresponded to a spinel structure at 2θ = (31°, 37°, 43°, 57° and 65°), corresponding to (220, 311, 400, 511 and 440) [20,29,30]. The spinel structure peaks were more intense and sharp for magnetite; on the other hand, the intensity and sharpness decreased by increasing the rGO loading (wt%). This can be attributed to the decrease in crystallinity for the composite as a result of increasing the rGO content over the content of magnetite.

The crystallinity of the prepared composites can be indicated by Scherrer’s equation, as shown in Equation (6):L = k λ/(β cosθ)(6)
where (L) is the crystalline domain size (nm), which is calculated according to the highest intense crystalline peak; (k) is the shape factor of the average crystallite (~0.9); (λ) is the wavelength in nm; and (β) is the full width have maximum in radians [31]. Table 2 summarizes the crystalline domain size (nm) for each composite. It is clear that the highest crystalline domain size was observed for Fe_3_O_4_ = 29.8 nm and this value decreased by increasing the rGO content until it reached 5.53 nm. This attitude reveals the clear impact of an increase in the rGO loading content on lowering the crystallinity/order of the nanocomposites.

The XRD data of the spinels (i.e., maghemite, magnetite, or other metal oxides) exhibited highly similar diffraction patterns as they all have the same crystalline structure and to shed light on the nature of the spinels, the powder of the samples was characterized using Raman spectroscopy. This technique is a powerful tool to provide the structural properties of materials and can offer unique information when analyzing nanostructured transition metal oxides as it registers the different vibrations due to the different cationic arrangements [29,32]. In particular, bearing in mind that the iron oxide phases forming part of the Fe_3_O_4_ clusters are prone to oxidation, we employed an excitation wavelength of 633 nm but with low power to achieve an appropriate penetration depth while avoiding any possible oxidation or even phase transition in the clusters.

Figure 3a illustrates the Raman spectrum for the prepared magnetite cluster nanoparticles. The peaks marked with the dashed black lines refer to the vibrational modes of Fe_3_O_4_ magnetite and the red ones represent maghemite (γFe_2_O_3_). It is clear that the main five peaks for magnetite appeared at 196, 306, 460, 538, and 668 cm^−1^, which represents the corresponding modes of magnetite as T_2g_ (1), E_g_, T_2g_ (2), T_2g_ (3), and A_1g_, respectively. On the other hand, the main three peaks of maghemite appeared at 350, 500, and 700 cm^−1^, which represents the corresponding modes of maghemite as T_2g_, E_g_, and A_1g_, respectively [29,33,34]. In this context, other characteristic features of the materials such as the black coloration of its powders or their strong response to external magnetic fields should be considered. Figure 3b shows the Raman spectra for the rGO and Fe_3_O_4_@rGO composites (MR5, MR10, MR20, and MR30). For the Fe_3_O_4_@rGO composites, the A_1g_ mode displayed at wave number 668 cm^−1^ confirmed the formation of magnetite in the acquired hybrid nanocomposites [35]; in the line with this, D and G bands can be easily distinguished for the rGO and Fe_3_O_4_@rGO composites. The in-plane breathing vibration of the sp^2^ carbon rings is attributed to the G band recorded at a wavenumber of 1590 cm^−1^ for rGO [36]. The D band is produced by the sp^2^ carbon lattice with sp^3^ deficiencies breathing out of the plane, which was detected at 1364 cm^−1^ for rGO [37]. It was well-noted that the formations of magnetite in the hybrid nanocomposites could reduce the intensity of the G and D bands. In addition, the obtained increased shift for the remarked G and D bands toward the left (decreasing) could be associated with the increase in Fe_3_O_4_ content in the conducted nanocomposites [38]. This could be ascribed to the Fe_3_O_4_ nanoparticles placing the position of carbon atoms in the rGO sheet and presenting defects in it [38]. Additionally, due to this phenomenon, the intensity ratios between the G and D bands (I_G_/I_D_) were 1.07, 0.99, 0.98, 0.97, and 0.96 for rGO, MR30, MR20, MR10, and MR5, respectively, as shown in Figure 4. It is clear that the value of I_G_/I_D_ was greater than (1) only for rGO (lower defects) and for all the other Fe_3_O_4_/rGO composites, the values were lower than (1), which indicates that the defects increased in the composites as the combination of Fe_3_O_4_ with the rGO sheets increased [39,40].

The magnetic characteristics of Fe_3_O_4_, MR5, MR10, MR20, and MR30 were studied using a vibrating sample magnetometer (VSM). Figure 5 shows the hysteresis loops of the Fe_3_O_4_ and Fe_3_O_4_@rGO composite materials measured at 298 K. Table 3 summarizes the VSM characteristics values of the developed samples under study. In this regard, it is clearly understood that the magnetic behavior for all of the prepared samples is from soft magnetic materials due to the narrow hysteresis loops [38,41]. Fe_3_O_4_ demonstrates the highest saturation magnetization (M_s_) with a value of 48.274 emu/g. The values of Ms decreased by increasing the loading of the diamagnetic rGO to 36.754, 26.057, 16.971, and 10.662 emu/g for MR5, MR10, MR20, and MR30, respectively. This behavior indicates the successful fabrication of Fe_3_O_4_@rGO composites [42]. Furthermore, the magnetic remanence (M_r_) value decreases as the percentage of the magnetite nanoparticles increases. The M_r_ value for Fe_3_O_4_ was 2.271 emu/g, and this value decreased to 0.994, 0.716, 0.427, and 0.236 emu/g for MR5, MR10, MR20, and MR30, respectively. The magnetite nanoparticles and Fe_3_O_4_@rGO composites exhibited lower values of coercivity (˂100) which reduced the effect of the reversible action in the hysteresis loops. Therefore, the developed Fe_3_O_4_ and Fe_3_O_4_@rGO composites could be considered as superparamagnetic materials [38,43] and play a vital role in EMI SE, which will be discussed later in detail.

Figure 6 illustrates the FTIR spectra for PU foam (neat) and P_3_MR20 in such a way as to know and analyze the type of interaction between the Fe_3_O_4_@rGO (filler) and PU foam (matrix). It can be clearly seen that all of the characteristic peaks of PU foam were well-determined. The stretching vibration of hydrogen bonding appeared at 3427 cm^−1^ and the unbonded (N–H) groups at 3520 cm^−1^. The (symmetric and non-symmetric) vibrations of the CH_2_ groups could be identified at 2930 and 2866 cm^−1^, respectively. The vibrational stretching for (C=O) appeared at 1722 cm^−1^. Additionally, the vibrational stretching of (C–O) could be recognized at 1228 cm^−1^. The non-symmetric stretching of the (C–O–C) peak was detected at 1072 cm^−1^. The peak of (N–H) distortion was detected at 1533 cm^−1^. Furthermore, the stretching vibrational peak of (C–N) could be identified at 1228 cm^−1^ [44]. The imine (C=NH) peak appeared at 1600 cm^−1^. By comparing the PU standard peaks and the detected FTIR spectra, it is clear that PU and P_3_MR20 were successfully prepared. The same peaks were also detected for both PU and P_3_MR20 without any changes. This indicates that the PU foam segments did not reveal any chemical reaction with Fe_3_O_4_@rGO (filler). There was a slight position shift that was recognized at 924, 1690, and 3300 cm^−1^. This may be attributed to a physical interaction due to the loan pair of electrons in such remaining oxygenated groups (in magnetite or rGO) and PU foam [45]. Interestingly, FTIR indicates that the interaction between Fe_3_O_4_@rGO and PU is physical interaction.

Figure 7 shows the SEM micrographs for (a) PU, (b) P_1_MR20, (c) P_2_MR20, and (d) P_3_MR20. For PU, the foam structure was clear and cylindrical cells with spherical shapes formed. The average cell size for PU was 317 µm. In Figure 7b, c, for P_1_MR20 and P_2_MR20, the average cell size was 210 µm and 95 µm, respectively. This indicates that when increasing the filler loading, the average cell size of the foam composite decreases. This phenomenon can be explained by the role of the Fe_3_O_4_@rGO composite as a filler in the foaming process of PU. When the loading of filler increases, the nucleation stage in the foaming process also increases. A large number of the formed cell nuclei consume a large amount of the evolved gas during the foaming process and decrease the expansion of the composite foam [46]. Furthermore, Fe_3_O_4_@rGO, as a filler, converted the composite foam cell structure from the closed cell, as shown in Figure 7a, to an open cell structure, as shown in Figure 7b,c, by damaging the cell walls and merging the cells. The Fe_3_O_4_@rGO filler was well-distributed in the cell walls and the solid part of the foam composite material, which indicates the homogeneity of the filler distribution in the preparation step. The cell density can be calculated from Equation (7) [47]:N_f_ = [nM^2^/A]^3/2^(7)
where N_f_ is the cell density (cells/cm^3^); n is the number of cells; A is the area of the SEM image (cm^2^); and M is the magnification factor.

The cell density for neat PU was 3.2 × 10^9^ cells/cm^3^, whereas for the other composite foams, it was 1.1 × 10^10^ and 3.5 × 10^10^ cells/cm^3^ for P_1_MR20 and P_2_MR20, respectively. It was noticed that the cell density increased when the filler loading increased. This could be explained by the increased rate of nucleation in the foam structure and an increase in the number of formed cells, which also increased the cell density, as shown in Figure 7a–c [47]. From Figure 7d, at a high filler loading, it is clear that the foam structure began to be damaged. The image displayed large linked holes and thick struts. This could be attributed to the large filler content increasing the viscosity of the solution in the preparation step, which disables the evolved gas from building in the foam structure and escape.

Figure 8 displays the EDX results for (a) P_1_MR20, (b) P_2_MR20, and (c) P_3_MR20. For the three prepared composite foam materials, the mapping of EDX indicates a good distribution of C, O, and Fe in the composite foam. This indicates the role of sonication and good dispersion of the filler (Fe_3_O_4_@rGO) in the polyol. Furthermore, the mass percentage of iron was increased by increasing the filler loading to 0.62, 3.21, and 8.41% for P_1_MR20, P_2_MR20, and P_3_MR20, respectively. This indicates the success in the preparation of the Fe_3_O_4_@rGO and Fe_3_O_4_@rGO/PU composite foam materials with different filler loading percentages.

The effect of filler loading on the thermal stability of the neat PU and PU composite foam materials is illustrated in Figure 9 and Table 4. Figure 9a shows that normal decomposition occurred for both PU and PU composite foams while Figure 9b shows the thermal decomposition reaction rate. In Figure 9, it can be seen that when increasing the temperature from room temperature to about 150 °C, the main weight loss percentage (3%) can be attributed to the evaporation of trapped gases and low molecular weight fragments. When the temperature increased to about 250 °C, the decomposition process occurred in the PU segments as shown in Equation (8):

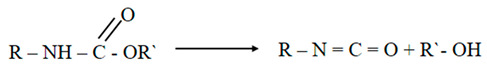
(8)

In this step, isocyanates and polyol were formed due to the decomposition of chemical bonding. These compounds are highly reactive and can be polymerized, forming diisocyanate and polyol decomposes, as shown in Equations (9) and (10) [48]:

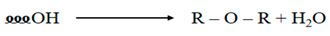
(9)

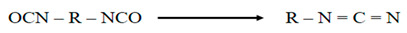
(10)

With the continuous increase in temperature (320–400 °C), the remaining polyols were decomposed to aliphatic ethers, CO and CO_2_. At higher temperatures, the produced secondary amines and gas evaporation continued, as shown in Equation (11).

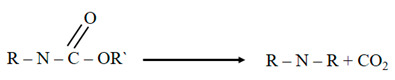
(11)

These thermal decomposition steps in the PU foam composites shifted to higher temperatures with an increase in the filler loading. This behavior occurs even though the filler has high thermal conductivity (5300 W/m.K).

This shift to a higher temperature may be attributed to the bonding between the filler and OH and CN groups in the PU segments, which delayed the decomposition temperature. In high filler loading (35%), the filler particles became closer and the thermal conductivity effect overcame the bonding between the filler and PU segments, and consequently, the thermal stability decreased [19].

Figure 10a demonstrates the stress–strain curves for PU, P_1_MR20, P_2_MR20, and P_3_MR20, which are relevant to the mechanical characteristics, mainly compressive modulus and compressive strength for the neat foam samples with different filler loadings. The curves can be divided into three main regions: (i) the linear region, where the slope of the curve is calculated and represents the compressive modulus; (ii) the plateau region, where the first point before the plateau that connects the linear region and plateau is determined and represents compressive strength; (iii) the densification region, where a rapid failure to the material occurs [49]. The compression results for the neat PU showed that the values of the compression modulus and compression strength were 0.54 and 5.62 MPa, respectively. Further addition of the filler enhanced the compressive modulus and strength to 4.44 and 12.22 MPa for P_1_MR20 and 5.03 and 15.6 MPa for P_2_MR20. The further addition of the filler deteriorated the mechanical behavior of the PU composite foams as the values of the compressive modulus and strength for P_3_MR20 were 0.53 and 5.11 MPa, respectively. So, the effect of filler loading on the mechanical characteristics (compressive modulus and compressive strength) of the PU composite foam materials, can be defined as the gain (G%). Gain can be calculated according to the results of neat PU from Equation (12) [49]:G = {[M_PU/Fe3O4@RGO_ − M_PU_]/M_PU_} × 100(12)
where G is the gain (%); M_PU_ is the compression modulus (strength) of a neat PU sample (MPa); and M_PU/Fe3O4@RGO_ is the compression modulus (strength) of the Fe_3_O_4_@rGO/PU samples (MPa). Figure 10b shows the effect of filler loading on the compressive modulus/strength and the gain. The addition of Fe_3_O_4_@rGO to the PU foam matrix enhanced the mechanical characteristics and the gain to a certain extent (25 wt%). The gain values increased from 722% and 117% (compressive modulus/strength) for P_1_MR20 to 831% and 174% (compressive modulus/strength) for P_2_MR20. Finally, the gain values decreased to reach −1.85% and −9.01% (compressive modulus/strength) for P_3_MR20. From the previous results, it is clear that Fe_3_O_4_@rGO acts as a reinforcing material that enhances the mechanical characteristics of the composite polymer foam until certain loading (25 wt%) [49]. When extra loading is introduced to the PU matrix, the large concentration of filler decreases the foam ability of the polymer composite and forms agglomerates with low dispersibility in the matrix. This phenomenon can be explained as follows. In the foaming process with high filler concentration, two regions are formed: (i) a heavily loaded region (due to filler agglomeration) and (ii) a lighter loaded region. The gases that formed during the foaming process prefer to escape from the heavily loaded region to the lighter one. This causes large holes and thick struts, as indicated in Figure 7d, which has a negative effect on the mechanical properties of the composite foam materials. Table 5 summarizes the compression test results.

The performance of the PU composite foam materials in EMI SE can be highly affected by two main factors: First, the ratio between Fe_3_O_4_ and rGO in the prepared filler Fe_3_O_4_@rGO, and the second factor is the concentration of Fe_3_O_4_@rGO as a filler in the PU foam matrix. Therefore, the optimum composition of the polymer foam composite from the point of view of SE can be determined by executing two sets of experiments. The first one has different filler compositions based on varying the ratios of Fe_3_O_4_/rGO and the same filler concentration in the PU matrix. In the second one, the best composition in Fe_3_O_4_@rGO as a filler was chosen by varying its loading content in the PU matrix to better determine the more efficient filler loading percentage. Figure 11 shows the SE results and the electrical conductivity measurements of PMR5, PMR10, PMR20, and PMR30 (the same filler loading in PU matrix with 35% and varying filler composition). Figure 11a illustrates the relation between reflection loss (RL) and frequency (8–12 GHz). For PMR5 and PMR10, the RL increased by increasing the frequency. The minimum RL values were −5.8 and −4.1 dB at 8 GHz and reached −9 and −6.1 at 12 GHz for PMR5 and PMR10, respectively. For PMR20 and PMR30, there was nearly a plateau noticed, with the values of −10 and −3.6 dB, respectively. The frequency dependency might be caused by structural factors such as the geometrical dispersion of the filler and the connection of electromagnetic waves with Fe_3_O_4_@rGO [50]. Furthermore, it was also noted that the higher performance in RL was achieved by PMR20, which indicates the synergistic effect between Fe_3_O_4_ and rGO at this composition (20% rGO:80% Fe_3_O_4_). Figure 11b illustrates the transmission loss (TL) as a function of the frequency (8–12 GHz). It is clear that for PMR5 and PMR10, TL decreased with increasing frequency. The maximum values of TL were −5 and −11.9 dB at 8 GHz and reached −3.5 and −9.8 at 12 GHz for PMR5 and PMR10, respectively. Furthermore, for PMR20 and PMR30, the values of TL increased with increasing frequency. The minimum values of TL were −20 and −24.6 dB at 8 GHz and reached −22.1 and −25.3 at 12 GHz for PMR20 and PMR30, respectively. This could be attributed to the increase in the RGO content in Fe_3_O_4_@rGO composite, which led to an increase in the electrical conductivity, as shown in Figure 11d, further leading to an increase in the TL values [51]. Additionally, a higher performance in TL was observed in PMR30 and this might be attributed to the higher content of rGO, and hence higher electrical conductivity levels. Figure 11c shows the total shielding efficiency (SE_Total_) as a function of the frequency (8–12 GHz). The minimum values of SE_Total_ were −10.9, −16.5, −30, and −28 dB at 8 GHz and increased to −12.4, −17, −33, and −30 dB for PMR5, PMR10, PMR20, and PMR30, respectively. It was noted that PMR20 demonstrated the best performance in SE_Total_ with average value of about −31 dB over the frequency band. This could be attributed to the synergistic effect of Fe_3_O_4_ and RGO in this composition, which achieved acceptable values in the field of SE [1], while in other fillers (PMR5, PMR10, and PMR30), the increase in rGO content hindered the effect of Fe_3_O_4_, which was responsible for the reflection losses. From the previous discussion, MR20 was selected as the best filler composition.

The optimum loading of MR20 in the PU foam matrix from the SE point of view was studied. The loading percentages (wt%) were selected according to some preliminary experiments and the maximum loading of the PU foam matrix. Figure 12 illustrates the SE parameters and the electrical conductivity results of P_1_MR20, P_2_MR20, and P_3_MR20. Figure 12a shows the RL as a function of the frequency (8–12 GHz). It was observed that there was nearly a plateau with values of −6.3 and −10 dB for P_2_MR20 and P_3_MR20, respectively. For P_1_MR20, the RL values increased from −3.2 dB at 8 GHz to −4.1 dB at 12 GHz. Furthermore, the increase in filler loading in the PU matrix increased the average RL values along with the frequency band, as shown in Figure 13a.This might be attributed to the increase in the super-paramagnetic Fe_3_O_4_@rGO loading percentage in the composite foam, which increased the magnetic losses, as discussed in the VSM results. Figure 12b shows the TL as a function of frequency (8–12 GHz). For P_1_MR20 and P_2_MR20, the maximum values of TL were −9.8 and −17.5 dB at 8 GHz and decreased to −5.4 and −14.8 dB at 12 GHz, respectively. For P_3_MR20, the minimum TL value was −20 dB at 8 GHz and this value increased to reach −23 dB at 12 GHz. It was also noted that the average values for TL increased by increasing the filler loading, as shown in Figure 12b. This could be attributed to the increases in the electrical conductivity in the PU composite foam due to the increase in the rGO content, as shown in Figure 12d [51]. Figure 12c illustrates SE_Total_ as a function of the frequency (8–12 GHz). It is clear that as the filler loading increased in the matrix, the total SE increased. The higher filler loading in P_3_MR20 demonstrated a higher SE_Total_ with an average value of −33 dB and this value decreased by decreasing the filler percentage in the matrix to reach −23.2 and −13 dB for P_2_MR20 and P_1_MR20, respectively. This behavior can be attributed to some factors that have a direct effect on the total shielding. First, with a higher Fe_3_O_4_@rGO content in the matrix, the particles of rGO become closer to each other and form electrically conductive paths that interact with the EM waves and increase the dielectric losses [14]. Furthermore, there are a large number of super-paramagnetic magnetite nanoparticles that form a strong magnetic field that interacts with the EM waves to generate the magnetic losses [52]. Finally, at higher filler loading, the cell density increases as indicated by SEM, at which the multiple reflections increase, which increases the total SE and maximizes the role of the porous structure. Table 6 shows a comparison between this work and the recently published literature.

Shielding efficacy and mechanical properties are two crucial aspects that assess the performance and determine the possibility of employing the developed materials in shielding applications. The loading percentage of the filler in the matrix plays a vital role, which has an extreme effect on both the SE and mechanical characteristics of the prepared shielding material. It was reported that increasing the filler loading enhances the SE, but also deteriorates the mechanical characteristics of the prepared composite materials, especially in foam matrices. A large amount of the published research has studied how to enhance the SE, but has ignored an important aspect, which is the mechanical properties. Figure 14 illustrates the optimization study among the SE, filler loading, and mechanical properties. It is clear that increasing the filler loading raised the SE to reach its maximum value of −33 dB at 35% Fe_3_O_4_@rGO. Additionally, the same attitude was noticed in the behavior of the mechanical properties (compressive modulus and compressive strength), but to a certain filler loading. The compressive modulus/strength was increased by increasing the filler loading to reach their maximum values of 5.3 and 15.6 MPa, respectively at 25 wt%. Further increase in the filler loading decreased the compressive modulus/strength values to 0.53 and 5.11 MPa at 35 wt% filler loading. Therefore, a comprehensive view of all aspects that determines the possibility of using these composite foam materials in the field, determined that the 25 wt% filler loading is the optimum loading. At this filler loading percentage, the maximum mechanical properties were achieved, and an acceptable total SE value was also obtained at −23.2 dB.

## 4. Future Work

There are still many gaps in the field of EMI SE that require large efforts from scientists and engineers. According to this work, we recommend studying the problem of PU composite foam recycling as it is considered as an environmental problem that needs a rapid and green solution.

## 5. Conclusions

Hybrid nanocomposites based on magnetite nanoparticles (Fe_3_O_4_) coated with reduced graphene oxide (rGO) by considering different rGO mass ratios (Fe_3_O_4_@rGO) were successfully synthesized through a co-precipitation method as well as fully characterized with different essential instrumentations. The TEM micrographs indicate the successful formation of iron oxide nanoparticles grouped into clusters of controlled size, in addition to a uniform distribution of Fe_3_O_4_ nanoparticles on the rGO sheets, taking into consideration the different rGO mass loading.

The X-ray diffraction (XRD) patterns of the developed samples revealed that the crystallinity of the obtained composites decreased with increasing rGO wt% percentage. In addition, the Raman spectra confirmed the successful preparation of the Fe_3_O_4_@rGO hybrid nanocomposites as well as the Fe_3_O_4_ cluster nanoparticles, even though with a portion of maghemite Fe_2_O_3_.

The vibrating sample magnetometer (VSM) values assure that magnetite nanoparticles and Fe_3_O_4_@rGO composites with different rGO content can work as a superparamagnetic material with lower coercivity (H_c_) and magnetic remanence (M_r_). The results conclude that Fe_3_O_4_@rGO has no chemical impact on the segmented PU foam, however, it was limited to a physical interaction that was evidenced by Fourier transform infrared (FTIR) analysis. Scanning electron microscope (SEM) micrographs demonstrated that exceeding the filler loading not only decreased the cell size, but also increased the cell density of the acquired composite foam matrices. The study concluded that both the compressive modulus/strength were enhanced by increasing the filler loading to 25 wt% (maximum values) and further loading could deteriorate the mechanical properties. The attitude and the obtained values show that the 20 wt% Fe_3_O_4_:80 wt% rGO composition can be considered as the optimum effective filler in the conducted composite foams; in addition, taking into account 35 wt% filler loading in the PU foam matrix as the most efficient percentage, thus offering high shielding efficiency, particularly of −33 dB as the total shielding achievement (SE)_Total_. This study offers a novel optimization study among the filler loading (wt%), SE, and mechanical properties. The optimization results indicate that the optimum Fe_3_O_4_@rGO loading in the PU foam matrix was 25 wt%, which achieved the highest mechanical properties (compressive strength: 15.6 MPa and compressive modulus: 5.03 MPa) and an acceptable SE value of −23.2 dB.

## Figures and Tables

**Figure 1 nanomaterials-12-02805-f001:**
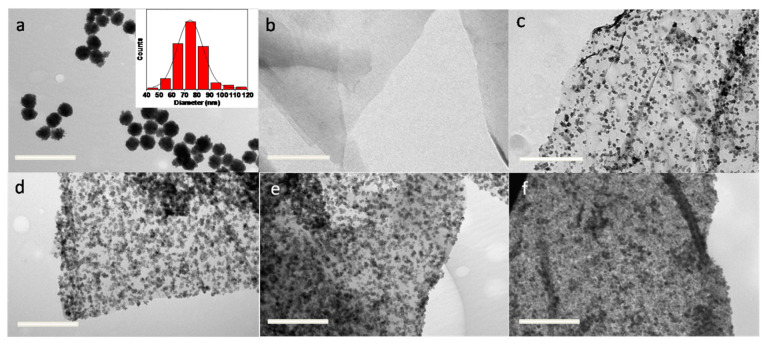
The TEM micrographs of the (**a**) clusters of Fe_3_O_4_/Fe_2_O_3_ nanoparticles, particle size distributions (inset), and (**b**) rGO sheets. Panels (**c**–**f**) corresponded to the TEM images of the Fe_3_O_4_/rGO hybrids as MR30, MR20, MR10, and MR5, respectively (scale bar: 500 nm).

**Figure 2 nanomaterials-12-02805-f002:**
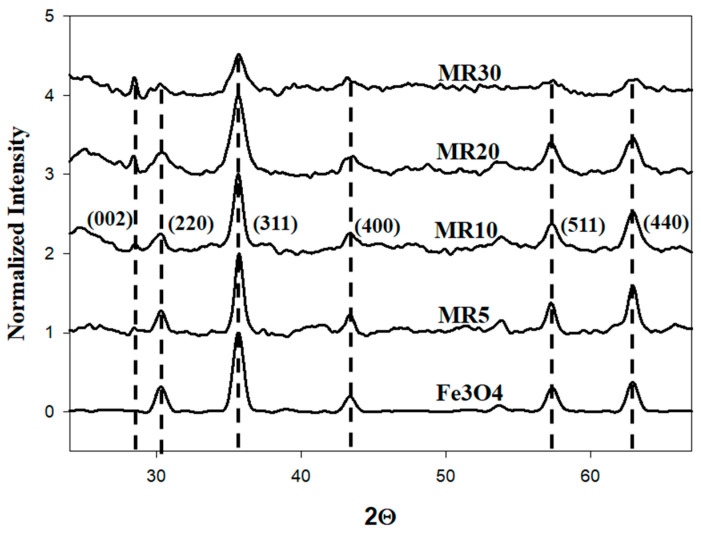
The XRD spectra for Fe_3_O_4_, MR5, MR10, MR20, and MR30.

**Figure 3 nanomaterials-12-02805-f003:**
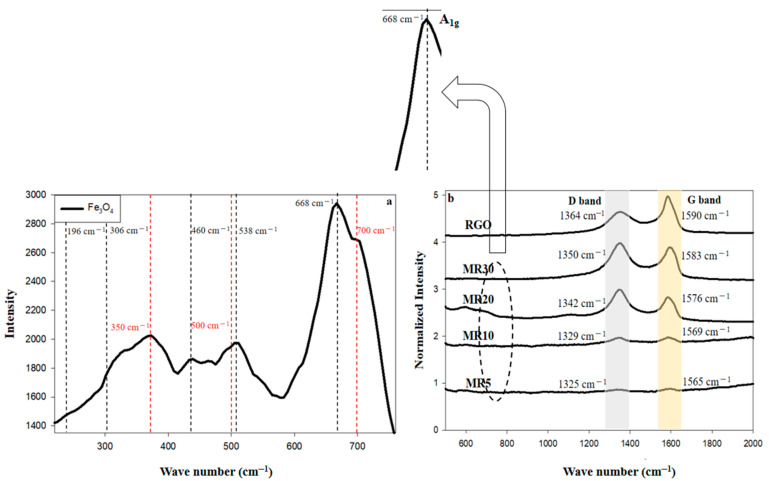
(**a**) The Raman spectrum for Fe_3_O_4_. (**b**) The Raman spectra for RGO, MR5, MR10, MR20 and MR30.

**Figure 4 nanomaterials-12-02805-f004:**
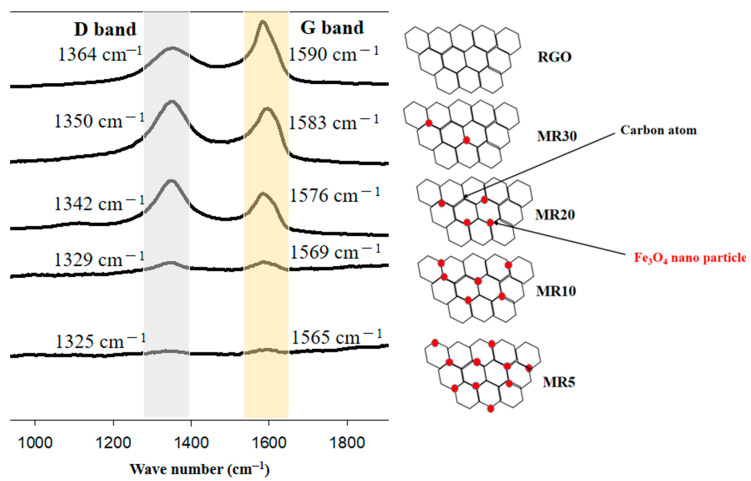
The interaction between rGO and Fe_3_O_4_ according to the Raman results.

**Figure 5 nanomaterials-12-02805-f005:**
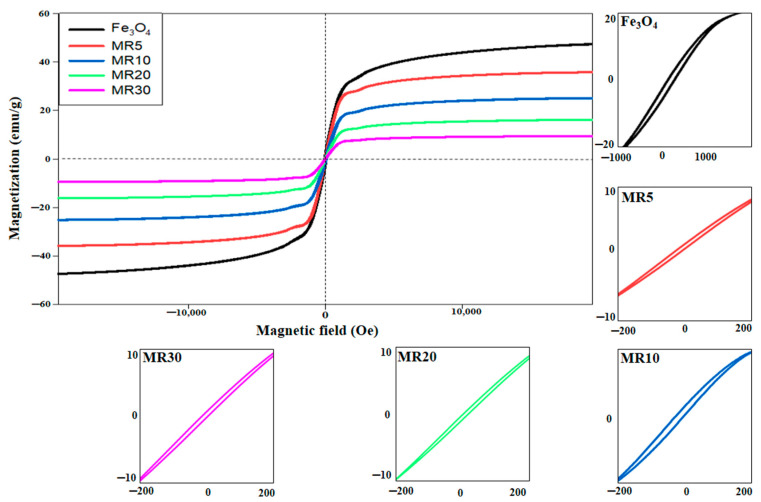
The hysteresis curves for Fe_3_O_4_, MR5, MR10, MR20, and MR30.

**Figure 6 nanomaterials-12-02805-f006:**
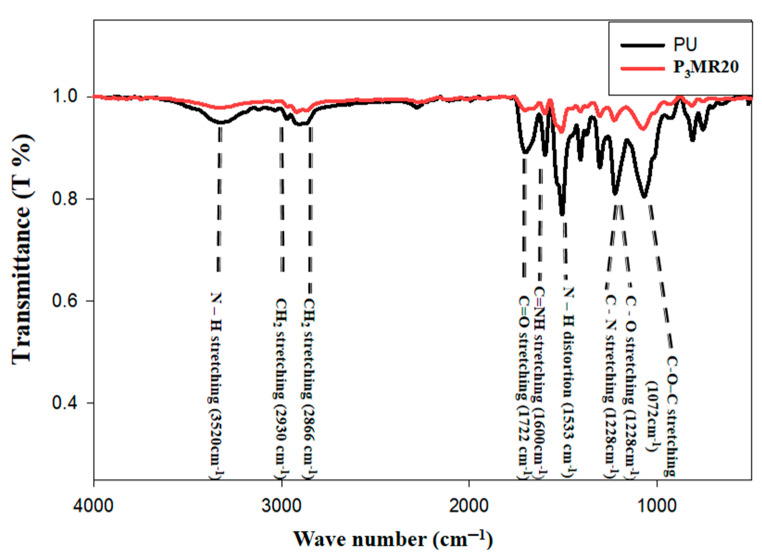
The FTIR spectra for PU and P_3_MR20.

**Figure 7 nanomaterials-12-02805-f007:**
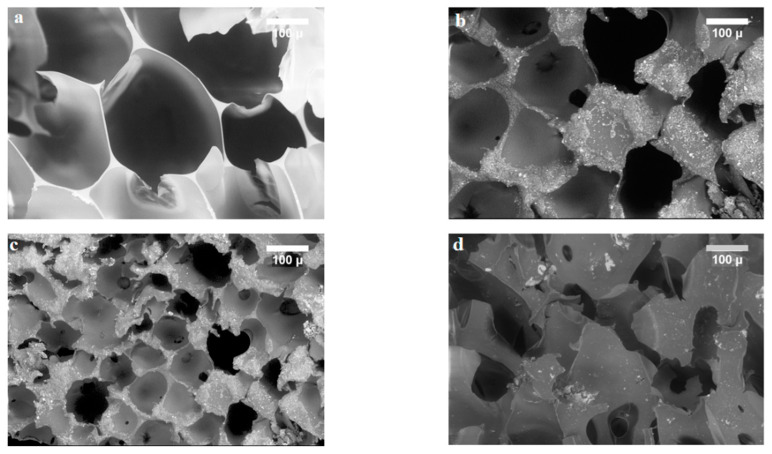
The SEM micrographs for (**a**) PU, (**b**) P_1_MR20, (**c**) P_2_MR20, and (**d**) P_3_MR20.

**Figure 8 nanomaterials-12-02805-f008:**
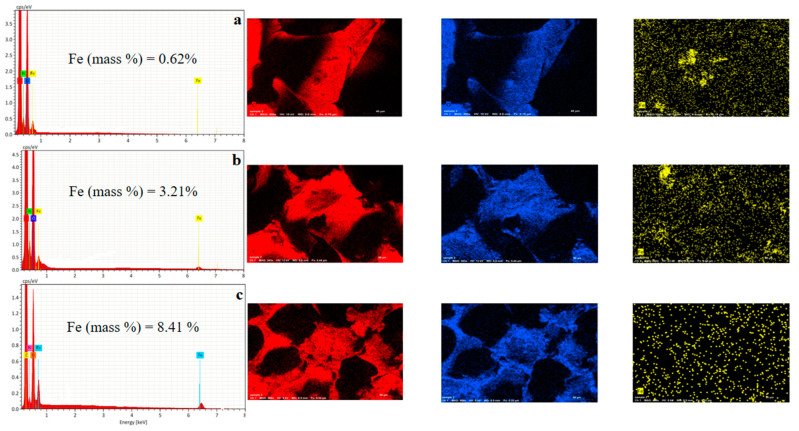
The EDX results and mapping for (**a**) P_1_MR20, (**b**) P_2_MR20, and (**c**) P_3_MR20.

**Figure 9 nanomaterials-12-02805-f009:**
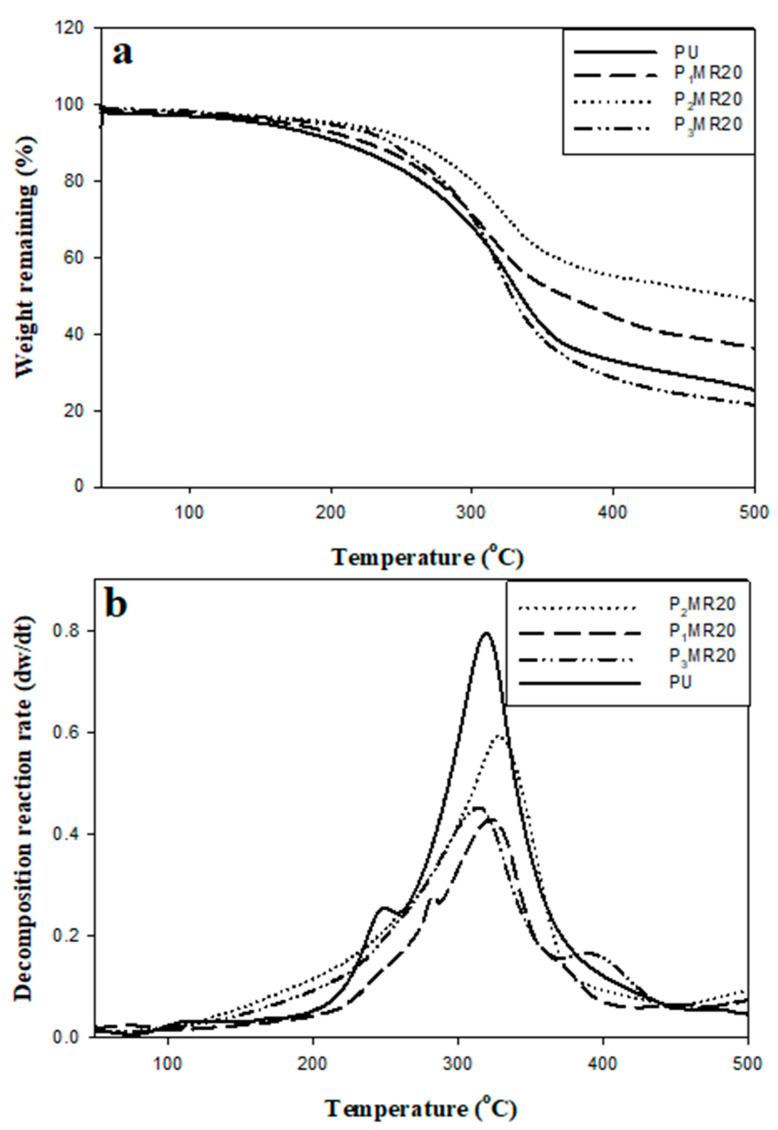
(**a**) The TGA thermograms and (**b**) the corresponding rate of the thermal decomposition reaction of PU-neat and Fe_3_O_4_@rGO/PU composites at different loading ratios.

**Figure 10 nanomaterials-12-02805-f010:**
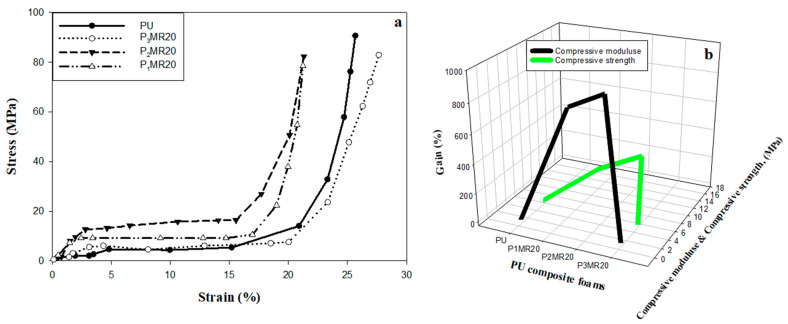
(**a**) Stress–strain curves for PU, P_1_MR20, P_2_MR20, and P_3_MR20. (**b**) Compressive test results for the PU composite foams.

**Figure 11 nanomaterials-12-02805-f011:**
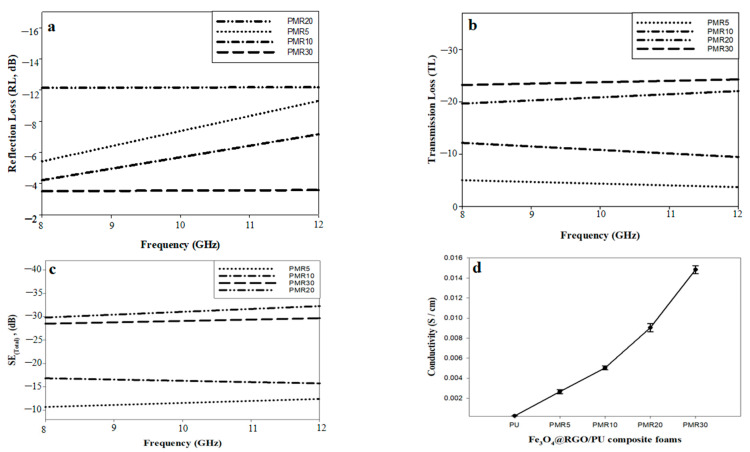
(**a**) Reflection loss. (**b**) Transmission loss. (**c**) Total shielding as a function of the frequency for PMR5, PMR10, PMR20, and PMR30. (**d**) The electrical conductivity results for the PU composite foams.

**Figure 12 nanomaterials-12-02805-f012:**
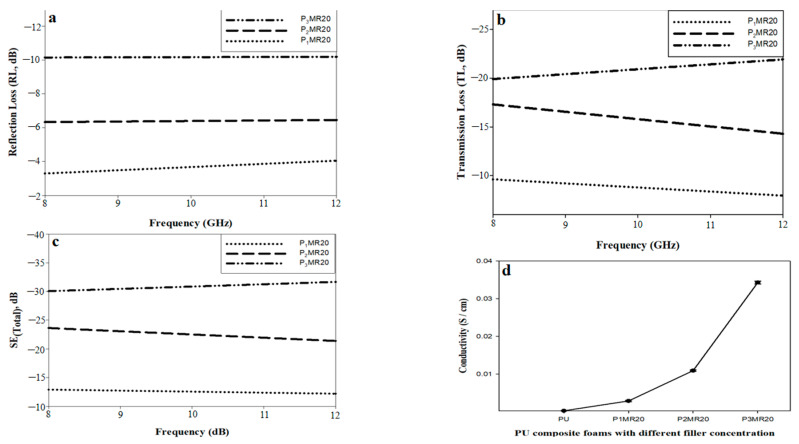
(**a**) Reflection loss, (**b**) Transmission loss, (**c**) Total shielding as a function of the frequency for P_1_MR20, P_2_MR20, and P_3_MR20. (**d**) The electrical conductivity results for the PU composite foams.

**Figure 13 nanomaterials-12-02805-f013:**
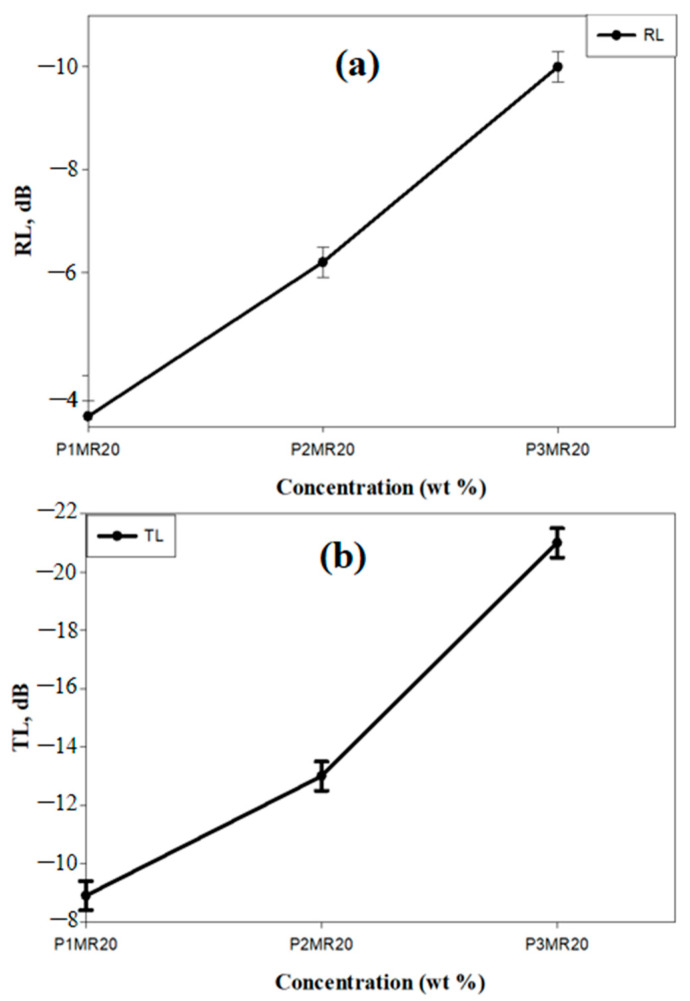
(**a**) Reflection loss and (**b**) transmission loss as a function of filler loading.

**Figure 14 nanomaterials-12-02805-f014:**
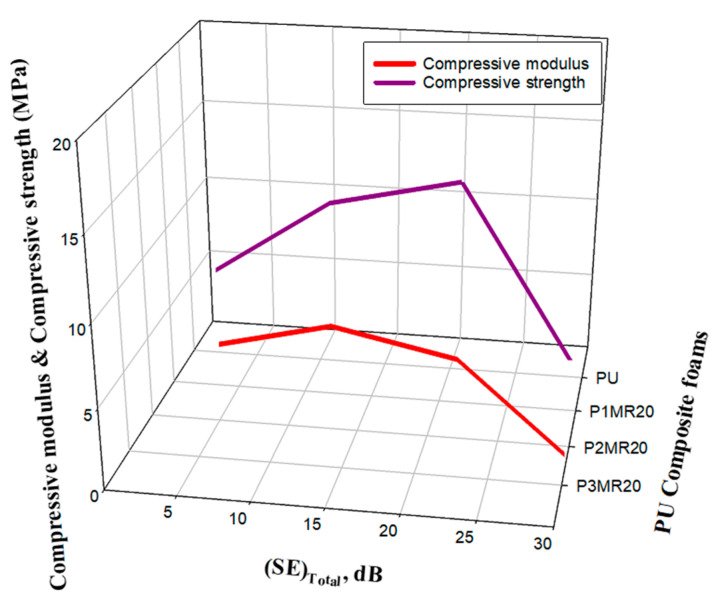
The optimization among the filler loading, SE, and mechanical properties.

**Table 1 nanomaterials-12-02805-t001:** The chemicals used in the preparation of the rGO, Fe_3_O_4_@rGO, and Fe_3_O_4_@rGO/PU composite foams.

Chemicals	Purity	Source
Graphite powder	99.5%	Haryana/India
Potassium permanganate	98%	Alpha Chemicals/India
Sulfuric acid	98%	Alpha Chemicals/India
Hydrochloric acid	36%	Alpha Chemicals/India
Hydrogen peroxide	35%	Alpha Chemicals/India
Ascorbic acid	98%	Alpha Chemicals/India
Ammonium ferrous sulfate hexahydrate	98.5%	Loba Chem/India
Ferric chloride anhydrous	98%	Loba Chem/India
Sodium hydroxide pellets	99.5%	Techno Pharm Chem/India
Ethanol	99.5%	Loba Chem/India
Polyol: Alcupol R-4520 (with a hydroxyl number of 343.5 mg KOH g)	-	Repsol Quimica/Spain
Methylene diphenyl diisocyanate (MDI) (viscous yellow liquid, viscosity 5260 cP at 25 °C, water content 2.3 wt%)	99.5%	Huntsman/Germany

**Table 2 nanomaterials-12-02805-t002:** The crystalline domain size (L, nm) for the different Fe_3_O_4_/rGO composites.

#	Composition	β (Radians)	L (nm)
1	Fe_3_O_4_	0.0049	29.8
2	MR5	0.0071	20.3
3	MR10	0.0108	13.5
4	MR20	0.0180	8.12
5	MR30	0.0264	5.53

**Table 3 nanomaterials-12-02805-t003:** A summary of the magnetic properties of the Fe_3_O_4_ and Fe_3_O_4_@RGO composites.

Magnetic Properties	Saturation Magnetization (M_s_, emu/g)	Magnetic Remanence (M_r_, emu/g)	Coercivity (H_c_, Oe)
Fe_3_O_4_	48.274	2.271	22.733
MR5	36.754	0.994	31.895
MR10	26.057	0.716	45.109
MR20	16.971	0.427	64.934
MR30	10.662	0.236	88.467

**Table 4 nanomaterials-12-02805-t004:** The weight loss (%) at various decomposition temperatures for the PU and Fe_3_O_4_@rGO/PU composite foam samples.

Sample Type	Weight Loss (%)					T_max_
	100 °C	200 °C	300 °C	400 °C	500 °C	
Neat PU	1.5	5.3	29.25	66.1	74.05	317
P1MR20	1.6	5.1	29.08	56.6	63.73	322
P_2_MR20	1.6	4.1	18.88	45.3	50.26	330
P_3_MR20	1.5	5	28.04	71.7	77.62	318

**Table 5 nanomaterials-12-02805-t005:** A summary of the compression test results.

Sample	Specimen	Compressive Modulus (MPa)			Compressive Strength (MPa)		
		Data	Mean	Gain (%)	Data	Mean	Gain (%)
PU	1 2 3	0.54 0.53 0.55	0.54	-	5.21 5.67 5.99	5.62	-
P_1_MR20	1 2 3	4.35 4.54 4.44	4.44	722 %	12.14 12.33 12.2	12.22	117%
P_2_MR20	1 2 3	5.03 5.02 5.04	5.03	831 %	15.5 15.7 15.8	15.6	174%
P_3_MR20	1 2 3	0.56 0.53 0.52	0.53	−1.85 %	5.11 5.1 5.13	5.11	−9.01%

**Table 6 nanomaterials-12-02805-t006:** A comparison between this work and the literature.

Matrix	Filler	Filler Loading (%)	Thickness (mm)	Conductivity (S/cm)	Freq. Range (GHz)	SE (dB)	Ref.
PVA	Fe_3_O_4_@rGO	35 wt%	3	˂0.001	8–12	−15	[53]
PVC		10 wt%	1.8	7.7 × 10^−6^		−13	[54]
PANI		66 wt%	3	2.6		−32	[18]
PS		2.24 v%	-	0.21		−30	[55]
PU		15 wt%	6	2.844 × 10^−3^		−13	This work
PU		25 wt%	6	10.9 × 10^−3^		−23.2	
PU		35 wt%	6	34.3 × 10^−3^		−33	

## Data Availability

The data that support the findings of this study are available from the corresponding authors, upon reasonable request.

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
