# Peer review of "Tuning Electro-Magnetic Interference Shielding Efficiency of Customized Polyurethane Composite Foams Taking Advantage of rGO/Fe3O4 Hybrid Nanocomposites"

_nanomaterials, 2022, doi:10.3390/nano12162805_

Round 1

Reviewer 1 Report

This paper developed a conducting Fe3O4@rGO/PU composite foam that with acceptable EMI SE and good mechanical properties. The reviewer has some issues as follows:

1、  p6, the inset in Fig.1a is not clear, please replace it.

2、  P6, it is clear that “the smallest” crystalline domain size was observed for Fe3O4 = 29.8 nm, the reviewer think it should be “the biggest”.

3、  p8, why the formation of magnetite can reduce the G and D bands’ intensity?

4、  P8, Fe3O4 nanoparticles will place the position of carbon atom in the rGO sheet and present defects in it, but in Fig.1, the least nanoparticles over the rGO sheets for MR5. Why the IG/ID of MR5 lower than other samples?

5、  p11, “Besides Fe3O4@rGO as a filler converted the composite foam cell structure from the “closed cell” as shown in figure7.a”, but the author also said “For PU the foam structure …with spherical openings are formed”, please explain.

Reviewer 2 Report

Authors present an incorporation of the metal oxide structures into the carbon-based materials as a filler of the polymer towards an application in the novel composites for enectro-magnetic field shielding.

This article covers new trends in nanomaterials, while some questions raise after reading proposed work. It is not clear how authors distinguish difference of the Fe3O4 and Fe2O3 mentioned in page 5. Proposed co-precipitation method can lead to the formation of both components while due to the 120C temperature, rather Fe2O3 can be obtained. The XRD studies only slightly differ for these two structures, so authors claim of the particlar clustuers of iron oxide (Fe3O4/-Fe2O3) has to be explained.

According to the synthesis, authors barely present the experimental conditions. Please deliver a detail information about the mass of each ingredient - step by step to make it easier to reader to navigate your procedure.

Please specify in the temperature in the magnetization results discussion in Kelvins. Please add the inset presentig narrow or lack of the hysteresis loop.

This work is full of editorial errors: lower case of the dot symbol, ml instead of mL, double spaces, and much more. Please correct them all. There are a lot of these errors at each page.

There is a lack of discussion in this work. Authors present conclusions, but the discussion would be valuable to this work. Please highlight the reason of the use of particular components in the discussion. There is too less comparison of the experimental results with the literature. Please compare your results with the recent literature. Please specify future perspectives of this work.

Round 2

Reviewer 1 Report

The reviewer think the manuscript can be accepted.